# 3D Biofilm Models Containing Multiple Species for Antimicrobial Testing of Wound Dressings

**DOI:** 10.3390/microorganisms10102027

**Published:** 2022-10-13

**Authors:** Kirsten Reddersen, Jörg Tittelbach, Cornelia Wiegand

**Affiliations:** Klinik für Hautkrankheiten, Universitätsklinikum Jena, 07743 Jena, Germany

**Keywords:** biofilm model, chronic wounds, antimicrobial testing, wound dressings, PHMB, silver

## Abstract

The treatment of chronic wounds presents a major challenge in medical care. In particular, the effective treatment of bacterial infections that occur in the form of biofilms is of crucial importance. To develop successful antibiofilm strategies for chronic wound treatment, biofilm models are needed that resemble the in vivo situation, are easy to handle, standardizable, and where results are readily transferable to the clinical situation. We established two 3D biofilm models to distinguish the effectiveness of wound dressings on important microorganisms present in chronic wounds. The first 3D biofilm model contains *Staphylococcus aureus*, *Escherichia coli*, and *Acinetobacter baumannii*, while the second is based on *Pseudomonas aeruginosa*. Bacteria are cultivated in a nutrient-rich agar/gelatin mix, into which air bubbles are incorporated. This results in a mature biofilm growing in clusters similar to its organization in chronic wounds. The models are convenient to use, have low variability and are easy to establish in the laboratory. Treatment with polihexanide and silver-containing wound dressings showed that the models are very well suited for antimicrobial testing and that they can detect differences in the efficacy of antimicrobial substances. Therefore, these models present valuable tools in the development of effective antibiofilm strategies in chronic wounds.

## 1. Introduction

The treatment of chronic wounds represents a major challenge in medical care [1,2]. In view of the increasingly aging population, treatment costs and efforts are expected to rise dramatically [3,4,5]. In chronic wounds, bacterial infections occur in the form of biofilms [4,6,7,8,9,10,11]. In biofilms, bacteria are aggregated and surrounded by extracellular matrix. This matrix contains polymeric substances such as polysaccharides, proteins, and DNA [11]. Through this shell, bacteria are very well protected from physical and chemical treatments and the body’s own defense mechanisms [7]. In addition to chronic wounds, biofilms are involved in several infectious diseases such as periodontitis, cystic fibrosis pneumonia, urinary tract infections, and indwelling medical device infections [11]. The effective treatment of biofilms in chronic wounds is of crucial importance to accelerate healing and reduce the burden on the patient as well as the cost of treatment. This can be achieved by using antibacterial dressings in addition to effective debridement [4]. A large number of these wound dressings are already on the market or in development [12,13,14,15]. To compare the effectiveness of different products and to develop successful antibiofilm strategies for treatment of chronic wounds with antimicrobial wound dressings, biofilm models are needed.

Ideally, biofilm models resemble the in vivo situation, are easy to handle, standardizable, and results are readily transferable to the clinical situation [16]. Several variables influence the formation of biofilms such as nutrient availability, hydrodynamic conditions, physicochemical properties of the substratum and type of microorganisms [11]. A number of existing biofilm models are reviewed in the literature [16,17,18,19,20,21]. These models differ in biofilm production and appearance in many factors, such as the surface or matrix used, the nutrients of the medium, the use of flow conditions, and the nutrient supply or use of monospecies or multispecies cultures, to name a few [17,19]. Biofilm models used for antimicrobial testing of wound dressings include 96-well plates [22,23,24], Centers for Disease Control (CDC) biofilm reactors [22,25], filter discs [22,26], flatbed perfusion [27], glass chamber slides [28], gauze implants on agar plates [29,30], or the Lubbock chronic wound biofilm (LCWB) on agar [31]. Unfortunately, there is no standardized methodological approach of biofilm models and the production of a biofilm that corresponds to the in vivo situation in chronic wounds remains a great challenge [32].

The objective of the present investigation was to establish 3D biofilm models for antimicrobial testing of wound dressings with multispecies communities of bacteria present in chronic wounds. *S. aureus*, *E. coli*, and *A. baumannii* were selected for the multispecies model, and *P. aeruginosa* for the monospecies model. One goal of our development was to ensure that the biofilm in the models does not grow predominantly on the surface, but rather as clusters in deeper layers of the models, which corresponds to the in vivo situation in chronic wounds [8,33]. Secondly, development of the models focused on the dimensions and haptic of the 3D biofilm models, resulting in environmental conditions during antimicrobial testing of the wound dressings that correspond to the clinical application. Finally, the newly developed models were tested to determine whether they are suitable for demonstrating the antimicrobial effects of wound dressings with different active antimicrobial compounds.

## 2. Materials and Methods

### 2.1. Preparation of 3D Biofilm Models

*Staphylococcus aureus* DSM 4910, *Escherichia coli* DSM 498, *Acinetobacter baumannii* DSM 102929, and *Pseudomonas aeruginosa* DSM 26644 (all DZMZ, Deutsche Sammlung von Mikroorganismen und Zellkulturen, Braunschweig, Germany) were cultivated on Columbia agar plates (Biomerieux, Nürtingen, Germany) overnight at 37 °C. Bacteria were suspended in Nutrient Rich Medium (NRM) consisting of 3 g/L lab lemco meat extract (Oxoid, Wesel, Germany), 10g/L special peptone (Oxoid, Wesel, Germany), and 5 g/L NaCl (Sigma-Aldrich, Taufkirchen, Germany) and cultivated for 3 h at 37 °C under vigorous shaking. Cultures were adjusted to an OD_600_ of 0.1 using 0.9% NaCl (Fresenius Kabi AG, Bad Homburg, Germany). Bacteriological agar (1.5%, Oxoid, Wesel, Germany) and gelatin (1%, Platin, 240 bloom, Roth, Karlsruhe, Germany) were dissolved in 60 mL NRM, autoclaved and stored at 4 °C until usage. Shortly before preparation of the 3D biofilm model, the agar/gelatin mixture was liquefied using a 95 °C water bath. During cooling of the mixture, two 50 mL Falcon tubes each with 4 stainless steel balls (3 mm diameter, Retsch GmbH, Haan, Germany) and 75 µL bacteria suspension were prepared. For the multispecies biofilm model, equal volumes of the OD_600_ 0.1 solutions of *S. aureus*, *E. coli*, and *A. baumannii* were mixed shortly before adding the agar/gelatin mixture. For the *P. aeruginosa* model, the OD_600_ 0.1 solution was used. The moment the agar/gelatin mixture had reached 43 °C, it was filled into the 50 mL falcons, after vigorous shaking resulting in many air bubbles the solution was united and shortly mixed. Then, 3 mL of this solution was transferred to each well of a 12-well plate (Greiner Bio-One, Frickenhausen, Germany). Plates were cooled for 30 min at 4 °C to accelerate solidification. Afterwards, the models were incubated for 48 h at 37 °C with 5% CO_2_. 

### 2.2. Visualization of the 3D Biofilm

After 48 h incubation, the mature 3D biofilm models were removed from the 12-well plate, halved and shock frozen using liquid nitrogen. After storage at −80 °C, freeze cuts with 4 µm thickness were mounted on slides. For visualization of the bacteria and the extracellular matrix, different staining techniques were applied. 

For staining of the bacterial cell walls, crystal violet (0.1% in distilled water, Sigma-Aldrich) was applied. DNA of the bacteria was visualized using DAPI (0.125 µg/mL in 0.9% NaCl, Sigma-Aldrich). Gram-staining was done following manufacturer’s instructions (Sigma-Aldrich). To distinguish internal bacterial DNA in living bacteria, and external DNA and DNA in dead bacteria, samples were stained using TOTO-1 iodide (2 µM in 0.9% NaCl, Invitrogen, Darmstadt, Germany) in combination with SYTO-60 (10 µM in 0.9% NaCl, Invitrogen). Staining was done 15–30 min in the dark. After rinsing with 0.9% NaCl, slides were covered in 0.9% NaCl and directly microscoped using the fluorescence microscope Axio Scope. A1 (Carl Zeiss AG, Jena, Germany) with appropriate filter blocs. Images were taken with the AxioCam MRc camera (Carl Zeiss AG, Jena, Germany) and analyzed using AxioVision 4.9 software (AxioVs40x64 V 4.9.1.0, Version 6.1.7601 SP 1 Build 7601, Carl Zeiss AG, Jena, Germany).

### 2.3. Stability Testing 

To investigate the stability of the 3D biofilm models, the multispecies 3D biofilm model and the 3D biofilm model with *P. aeruginosa* was probed shortly after production at day 0 and after 24 h, 48 h, and 72 h, and microbial count was determined. Models were divided and one-eighth of each was shredded in a 2mL reaction tube containing 4 stainless steel balls and 1 mL 0.9% NaCl in a vibration mill (MM301, Retsch GmbH) for 15 min. Samples were thoroughly vortexed and transferred with 4 × 1 mL 0.9% NaCl into a 15 mL Falcon tube. The microbial count of the samples was determined by serial dilution in 0.9% NaCl and plating on Columbia agar plates. After overnight incubation at 37 °C with 5% CO_2_, colonies were counted. Colonies in the multispecies model were distinguished based on their respective morphologies. *S. aureus* colonies were very small with a golden/white appearance. *A. baumannii* colonies were clearly white pigmented with a sharply defined edge. *E. coli* colonies were poorly pigmented with a blurred edge.

### 2.4. Incubation with Wound Dressings

To demonstrate the suitability of the 3D biofilm models for antimicrobial testing of wound dressings, the 48 h-old mature biofilms were incubated with different antimicrobial and non-antimicrobial commercial wound dressings for 24 h at 37 °C with 5% CO_2_. Two bacterial nanocellulose wound dressings (Suprasorb^®^ X, Lohmann&Rauscher, Suprasorb^®^ X + PHMB, Lohmann&Rauscher) and three alginate wound dressings (Suprasorb^®^ A, Lohmann&Rauscher, Suprasorb^®^ A + Ag, Lohmann&Rauscher, Acticoat Absorbent with SILCRYST^TM^, Smith&Nephew) were compared. Pieces Of each wound dressing with a diameter of 2.1 cm were prepared in an aseptic manner and fitted onto the models. The wound dressings were carefully pressed on to ensure good contact with the biofilm model. The alginate dressings were additionally loaded with 500 µL PBS (Bioconcept) to ensure moist conditions. Untreated controls were incubated in a humid chamber at 37 °C with 5% CO_2_.

### 2.5. Quantification of the Antimicrobial Effect of Wound Dressings

For the quantification of the antimicrobial effect after 24 h incubation with the wound dressings, the dressings were removed from the 3D biofilm models and the microbial count was determined as described before. 

### 2.6. Statistics

Experiments were performed in duplicate, and each sample was measured in two replicates. All values presented are expressed as means ± SD. One-way analysis of variance was carried out to determine statistical significances (Microsoft^®^ Excel 2016). Differences were considered statistically significant at a level of *p* ≤ 0.05 (*), *p* ≤ 0.01 (**) and *p* ≤ 0.001 (***).

## 3. Results

### 3.1. Establishment of the 3D Biofilm Model

For the establishment of the 3D biofilm model, different percentages of agar and gelatin were dissolved in nutrient-rich medium to produce a haptic that resembles chronic wounds and that is easy to handle in the laboratory. A mixture of 1.5% agar and 1% gelatin was found to be optimal. For the incorporation of cavities to produce bacterial clusters in the model that resemble the appearance of microbes in chronic wounds [8], air bubbles were produced in the 3D models by vigorous shaking of the liquid agar/gelatin mixture with stainless steel balls just before the liquid was poured into 12-well plates. In Figure 1, the distribution of the air bubbles in the solidified 3D model is shown in top view (A) and cross section (B). Using these optimized conditions, a multispecies 3D biofilm model was produced by addition of *S. aureus*, *E. coli*, and *A. baumannii*. After maturation for 48 h, clusters of biofilm are visible in the top view (Figure 1C). The incorporation of *P. aeruginosa* into the 3D biofilm model resulted in a green appearance of the model after 48 h maturation (Figure 1D).

### 3.2. Visualization of the 3D Biofilm

For the visualization of the 3D biofilm and the extracellular matrix, different staining techniques were applied. In Figure 2, microscopic images of the mature multispecies 3D biofilm with *S. aureus*, *E. coli*, and *A. baumannii* are shown. The peptidoglycans of the bacterial cell walls were stained with crystal violet. A very dense microbial population in the mature multispecies 3D biofilm is seen in Figure 2A. The bacteria used in this model are a mixture of Gram-positive *S. aureus* and Gram-negative *E. coli* and *A. baumannii*. The Gram-negative bacteria in this model differ in size and appearance. Gram staining of the mature multispecies 3D biofilm displayed that the bacteria are present in clusters of 20–100 µm diameter, with some of them located very close to each other. These bacterial clusters are not only present near the surface of the model, but also in deeper layers. This distribution of biofilm clusters resembles the appearance of microbial biofilms in chronic wounds [8]. In Figure 2B, cocci of *S. aureus* with dark purple color are located near rods of *E. coli* with reddish purple color. Figure 2C shows the co-localization of a cluster of dark purple cocci of *S. aureus* with a cluster of reddish rods of *A. baumannii*. The combination of the DNA stains TOTO-1/SYTO 60 enables on the one hand the distinction between living bacteria (red color) and dead bacteria (green color). On the other hand, external DNA is stained very sensitively. In Figure 2D–F, the vital centers of bacterial clusters are shown in red colors. Additionally, long green strands of external DNA are visible (Figure 2D,E) that form a part of the extracellular matrix of the mature biofilms. In Figure 2F, a cluster of cocci of *S. aureus* is in close proximity to a cluster of rods of *E. coli*, surrounded by a large amount of external DNA. 

In Figure 3, microscopic images of the mature 3D biofilm model with *P. aeruginosa* are shown. Crystal violet staining (Figure 3A) for the visualization of bacterial cell walls and DAPI staining (Figure 3B) for the detection of bacterial DNA display a very dense bacterial growth in the mature biofilms. Gram staining of the 3D biofilm model with *P. aeruginosa* confirms the clustered growth of rod-shaped bacteria (Figure 3C). Staining with TOTO-1/SYTO 60 of the 3D biofilm model with *P. aeruginosa* shows a very active center of clustered biofilm growth (Figure 3D). Additionally, long strands of external DNA are visible, which are part of the extracellular matrix of the mature biofilm. 

### 3.3. Stability of the 3D Biofilm Model

To investigate the robustness and stability of the developed multispecies 3D biofilm model with *S. aureus*, *E. coli*, and *A. baumannii* and the 3D biofilm model with *P. aeruginosa* over a longer time period, the models were probed every 24 h for up to 72 h and microbial count was determined. As seen in Figure 4A, the microbial count in the multispecies 3D biofilm model is stable from 24 h up to 72 h. Furthermore, it was observed that the ratio of bacteria to each other is stable and did not change in this period. Similar results were observed in the 3D biofilm model with *P. aeruginosa*, where the bacterial count was stable from 24 h up to 72 h (Figure 4B). 

### 3.4. Quantification of the Antimicrobial Effect of Wound Dressings

To demonstrate the suitability of the models for antimicrobial testing of wound dressings, the multispecies 3D biofilm model with *S. aureus*, *E. coli*, and *A. baumannii* and the 3D biofilm model with *P. aeruginosa* were incubated for 24 h with antimicrobial and non-antimicrobial wound dressings. The dressings were made from bacterial nanocellulose (SupX, SupX + PHMB) or alginate (SupA, SupA + Ag, Acticoat). The antimicrobial effects of polihexanide (SupX + PHMB) and silver (SupA + Ag, Acticoat) were compared (Figure 5A,B). In the multispecies 3D biofilm model, incubation with SupX + PHMB resulted in a significant reduction in all bacteria present in the model compared to the untreated control (Figure 5A). For *S. aureus*, a strong reduction in the bacterial count of more than 3 log levels was observed. This wound dressing was the only dressing with statistically significant antimicrobial effects on *S. aureus* in this study. Additionally, incubation with SupX + PHMB resulted in the strongest antimicrobial effects on *E. coli* and *A. baumannii* compared to the other tested antimicrobial wound dressings with silver. Incubation with the nanocellulose dressing without antimicrobial additives (SupX) resulted in no or little effects on the microbial count compared to the untreated control. Incubation of the multispecies 3D biofilm model with wound dressings containing silver ions and nanocrystalline silver reduced the bacterial count of *E. coli* and *A. baumannii* significantly compared to the untreated control. No differences in antimicrobial effect were observed between the two tested dressings SupA + Ag and Acticoat. For *S. aureus*, no significant reduction in the bacterial count was observed after incubation with silver containing wound dressings (Figure 5A). The alginate dressing without antimicrobial additives (SupA) showed no or little effect on bacterial counts in the multispecies 3D biofilm model compared to the untreated control. In the 3D *P. aeruginosa* biofilm model, incubation with the antimicrobial wound dressings SupX + PHMB, SupA + Ag, and Acticoat for 24 h resulted in a significant reduction in the bacterial count (Figure 5B). No differences in the antimicrobial effects were observed between SupX + PHMB and SupA + Ag. However, a significantly stronger antimicrobial effect was observed after incubation with Acticoat compared to SupA + Ag. No or little effect was apparent after incubation with the non-antimicrobial wound dressings SupX and SupA, with the effect of the alginate SupA on *P. aeruginosa* being comparable to the effect on *A. baumannii* in the multispecies model (Figure 5B). 

The testing of wound dressings in both newly developed 3D biofilm models showed very low variability in the results, demonstrating the robustness and standardizability of the biofilm models.

## 4. Discussion

In this study, two 3D biofilm models were developed containing multiple species that are commonly present in chronic wounds. The microbial population in the established models is stable for up to at least 72 h, recommending these models for antimicrobial testing of wound dressings.

The use of antimicrobial dressings in addition to effective wound debridement is a key factor in the consensus guidelines for treatment of biofilm in chronic nonhealing wounds [2]. For comparing the effectiveness of these wound dressings in fighting biofilms, models are needed that are easy to handle and standardizable with results readily transferable to the in vivo situation [16]. 

In our experiments, the bacteria used to establish the 3D biofilm models were selected based on those typically found in biofilms of chronic wounds [9,10,34,35,36]. The established model with a multispecies community included *S. aureus*, *E. coli*, and *A. baumannii*. Additionally, a 3D biofilm model with *P. aeruginosa* was designed. *S. aureus*, one of the most spread pathogens in chronic wounds, was identified in 71% of patients with chronic wounds, of which 30% carried methicillin-resistant *S. aureus* (MRSA) [36]. MRSA is one of the high priority microbes on the WHO list of antibiotic-resistant bacteria, a list that demands research in novel antibacterial strategies [37]. *E. coli* was found in 14% [10] or 5% [34] of chronic wound samples, respectively. As a commensal bacterium it resides in the gastrointestinal tract, but as an opportunistic pathogen, it can cause a variety of infections in the human body [38]. *A. baumannii* is an important nosocomial pathogen with an increasing resistance to antibiotics. Carbapenem resistant *A. baumannii* has the highest critical priority of all antibiotic resistant bacteria on the WHO list [37]. *A. baumannii* was identified in up to 5% of chronic wound samples [10]. *P. aeruginosa* is one of the most common opportunistic pathogens found in up to 35% of chronic wound samples [10]. The formation of biofilms is one of the main survival strategies of *P. aeruginosa* resulting in infections of, e.g., the skin or lung tissue in cystic fibrosis that are difficult to treat [39]. *P. aeruginosa* also has the highest critical priority on the WHO list [37]. 

The overwhelming majority of biofilm models established so far are based on surface attachment as a crucial component, which makes them highly suitable as models for foreign body infections [40]. Biofilms in these models often display a very high cell density with a thickness of several hundred micrometers or, e.g., a mushroom shape that was never observed in wound infections [40]. However, in chronic wounds, solid surfaces are absent, and biofilms display distinct small patches of clusters with diameters of 5–200 µm located not only near the surface, but also in the tissue [8,11].

In our study, 3D biofilm models with biofilm clusters were established that correspond very well to in vivo findings. Bacteria in our models are present in clusters of 20–100 µm diameter surrounded by extracellular matrix. These clusters are not only present near the surface, but also in deeper layers of the models that have an average height of 8 mm. The spatial nature of the models allows for good transferability to the in vivo situation during antimicrobial treatment with wound dressings, as effects on biofilm not only in direct contact, but also in deep tissue, can be recorded. So far, there are only a few in vitro biofilm models not based on surface attached biofilms growing in clusters [31,41,42,43] 

In more than 80% of chronic wound biofilms, two or more bacteria species are identified [20]. Despite the presence of various bacteria in chronic wounds, to date, a true multispecies biofilm has not been observed in wounds [40]. It is assumed that the species do not directly mix, but stay in their own ecological spaces. It has been postulated that, although different species do not necessarily interact, they all influence the host cells which leads to more adverse effects when multiple species are present in a wound [16]. In order to represent the in vivo situation of chronic wounds in biofilm models as closely as possible, it is necessary to establish models that contain different wound pathogens. In our study, a multispecies 3D biofilm model containing *S. aureus*, *E. coli*, and *A. baumannii* was successfully established. From 24 h up to at least 72 h, a stable population of all three bacteria was observed with a constant ratio to each other. Microscopic visualization demonstrated homogeneous clusters formed by the respective bacteria, with different bacteria in close proximity to each other. External DNA as part of the extracellular matrix of a mature biofilm was visualized using the DNA stains TOTO-1/SYTO60. This proof of maturation of the biofilm is important for comparability of the model to chronic wounds, since there is a different behavior of mature biofilms compared to growing biofilms [2]. In recent years, the need to develop biofilm models with more than one bacterial species has been recognized. Models exist that incorporate two different bacteria, e.g., *S. aureus* and *P. aeruginosa* in a multi-layered agar model [43], in a cellulose on agar model [44], or in a coagulated Wound-Like Medium model [45]. Furthermore, biofilm models with three [46,47,48], four [31], or five bacteria species [44] were established. The bacteria that were used in these models were primarily *S. aureus*, *P. aeruginosa*, *E. coli*, *Enterococcus faecalis*, *Streptococcus pyogenes*, and *Bacillus subtilis*. Antibiotic tolerance in these multispecies biofilm models was increased compared to monospecies biofilms [45]. This highlights the importance of establishing biofilm models containing multiple species to increase the translatability of in vitro antimicrobial testing to the in vivo situation.

To date, there are a lack of standardized methods for experimentally studying biofilms using clinically applicable in vitro models for antimicrobial testing of wound dressings [32]. This lack of regulatory guidance complicates the comparison of various studies on the effectiveness of different antimicrobial wound dressings. In the literature, there are a variety of biofilm models described that are used for antimicrobial testing of wound dressings. These models include microtiter plate models [22,24], CDC models [22,25], constant depth film fermentors [49], drip flow reactor models [46,50], models with gauze [29,30] or cellulose filters [22,26,44] on agar, layered agar models [43,51], or the LCWB on agar model [31]. However, there are several aspects in these models that complicate the transferability of the data generated by these biofilm models to the in vivo wound situation. Surface attachment of the biofilm is a crucial component in most of the established in vitro biofilm models, which certainly plays an important role in other areas of biofilm research, e.g., foreign body infections, but does not occur in chronic wounds. Furthermore, in models such as CDC or drip flow reactor, the biofilm is produced under high shear forces, which do not occur in chronic wounds. This leads to very different geometries of the produced biofilm in terms of thickness and shape compared to the in vivo situation [40].

In our study, the established 3D biofilm models containing multiple species and models containing *P. aeruginosa* were successfully used for the antimicrobial testing of wound dressings containing different active antimicrobial compounds such as PHMB, silver ions, and nanocrystalline silver. Using the biofilm model with *S. aureus*, *E. coli*, and *A. baumannii*, after 24 h incubation, an antimicrobial effect of the wound dressings with active compounds compared to the wound dressings without antimicrobial additives could be detected. The highest antimicrobial effects with significant reduction for all three bacteria compared to the untreated control was observed for Suprasorb X + PHMB with the highest effect on *S. aureus*. The antimicrobial effect of PHMB was proven in a number of in vitro and in vivo studies [52,53,54,55,56], and it is recommended as a first-choice agent for treatment of infected chronic wounds [57]. PHMB interacts with negatively charged phospholipids in the bacterial membranes leading to loss of integrity followed by cell death [54]. The observed antimicrobial effects of PHMB dressings in our established multiple species biofilm model are in accordance with previous in vitro investigations [42,43,53,55,58,59]. 

The incubation of multispecies 3D biofilm model with the antimicrobial wound dressings containing ionic silver and nanocrystalline silver had comparable effects and reduced the bacterial load of *E. coli* and *A. baumannii* significantly. The antimicrobial effect of silver for treating infections dates back to about 1000 BC, and nowadays, there is a variety of wound dressings with silver as active antimicrobial agent on the market [60]. The antibacterial effect of silver is based on the toxic effect on the respiratory enzymes, on the inhibition of protein synthesis as well as on the inhibition of bacterial DNA replication. In contrast to normal silver ions, the use of nanocrystalline silver extremely increases the surface area of the elemental silver, which leads to a higher concentration of released silver ions upon contact with wound exudate. The antimicrobial effect of silver containing wound dressings on biofilms was investigated in several studies [24,30,49,58,61]. However, the tests differ greatly with respect to the biofilm models used, their microbial composition and geometry, their aging, and the incubation time with the wound dressings, which makes it very difficult to compare the results of these studies and explains partly contradictory results. Confirming the results in our studies, Kostenko et al. 2010 [24] and Percival et al. 2007 [62] also observed antimicrobial effects of silver and nanocrystalline silver containing wound dressings on *E. coli* biofilms. 

The treatment of the *P. aeruginosa* 3D biofilm model with the wound dressings for 24 h resulted in a significant reduction in the bacterial load with all antimicrobial wound dressings compared to the wound dressings without active additives. These results are consistent with previous investigations [24,43,58,61]. A comparable antimicrobial effect was observed using the PHMB containing wound dressing and the dressing with ionic silver. However, the antimicrobial effect of the dressing with nanocrystalline silver was significantly stronger compared to the dressing with ionic silver. This observation was confirmed by Kostenko et al. 2010, who studied the effects of silver containing wound dressings on *P. aeruginosa* biofilms [24].

After 24 h treatment of the established 3D biofilm models with antimicrobial wound dressings, there was no complete eradication of the bacteria. This goes hand-in-hand with the recommendation of the consensus guideline for the treatment of chronic wounds that biofilms in chronic wounds should not be treated with antibacterial dressings alone, but always in combination with prior debridement [2]. In the literature, there is no target value in the log-reduction of antibiofilm agents that is necessary to cure an infection, since on the one hand, there are no standard biofilm models, and on the other hand, there are different factors that influence healing of a chronic wound such as the type of infection, the infecting strain, or the immune status of the patient [32]. The lack of target values and standardized methods makes it difficult for companies to register new anti-biofilm products or processes with a regulatory agency. This further emphasizes the need to establish appropriate biofilm models to improve transferability of benchside data to bedside treatment. 

## 5. Conclusions

In our investigations, we demonstrated the suitability of the newly developed 3D biofilm models containing multiple species to detect and compare the antimicrobial effects of various wound dressings with different active antimicrobial compounds.

## Figures and Tables

**Figure 1 microorganisms-10-02027-f001:**
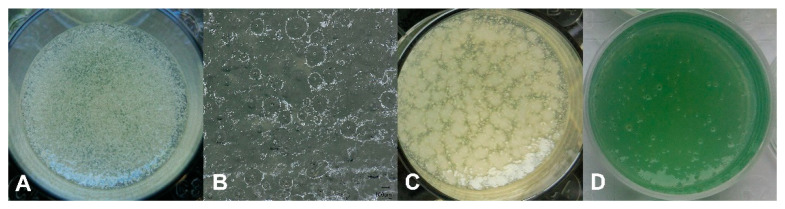
Incorporation of cavities into the agar/gelatin 3D-model without microorganisms (**A**,**B**) in (**A**) top view and (**B**) cross section. Top view of the mature 48 h multispecies 3D biofilm model with *S. aureus*, *E. coli*, and *A. baumannii* (**C**) and of the 3D biofilm model with *P. aeruginosa* (**D**).

**Figure 2 microorganisms-10-02027-f002:**
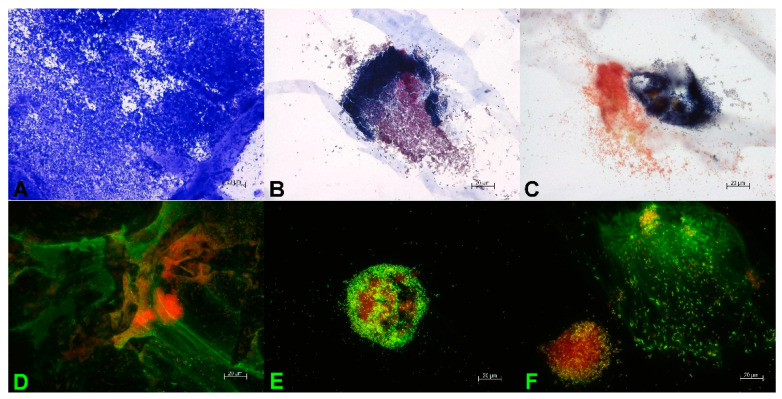
Microscopic evidence of the mature 48 h multispecies 3D biofilm with *S. aureus*, *E. coli*, and *A. baumannii*. (**A**) Staining with crystal violet confirms dense microbial population. Gram staining reveals a close proximity of clustered populations of the Gram-positive *S. aureus* (dark purple) and the Gram-negative *E. coli* (reddish purple) (**B**), or the Gram-negative *A. baumannii* (reddish) (**C**). Staining with TOTO-1/SYTO 60 (**D**–**F**) shows on the one hand the clustered growth of the microorganisms with a very vital, red center (**D**–**F**). On the other hand, the external DNA of the extracellular matrix is visible as long green DNA strands (**D**,**F**). In (**F**), the close proximity of a *S. aureus* cluster (left) and *E. coli* cluster (right, surrounded by large amounts of external DNA) is visible.

**Figure 3 microorganisms-10-02027-f003:**
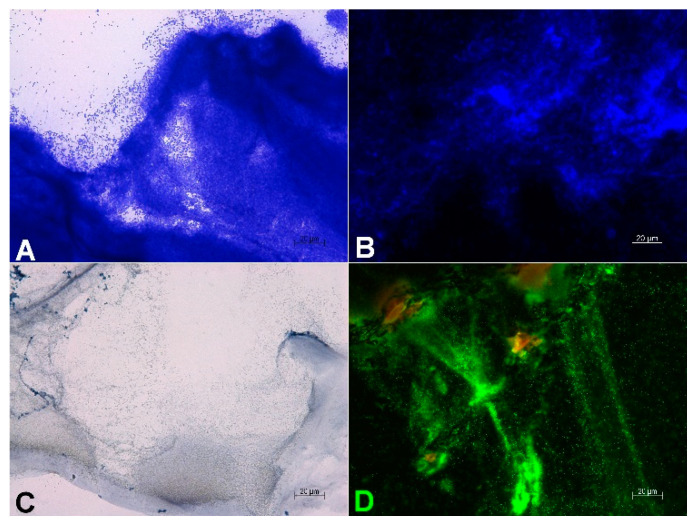
Microscopic evidence of the mature 48 h 3D biofilm with *P. aeruginosa*. Crystal violet staining (**A**) and DAPI staining (**B**) show a dense microbial population. With Gram-staining (**C**), clusters of Gram-negative rods are visible. TOTO-1/SYTO 60 staining (**D**) reveals vital, red centers of *P. aeruginosa* clusters and long strands of external DNA (green) as part of the extracellular matrix of the mature biofilm.

**Figure 4 microorganisms-10-02027-f004:**
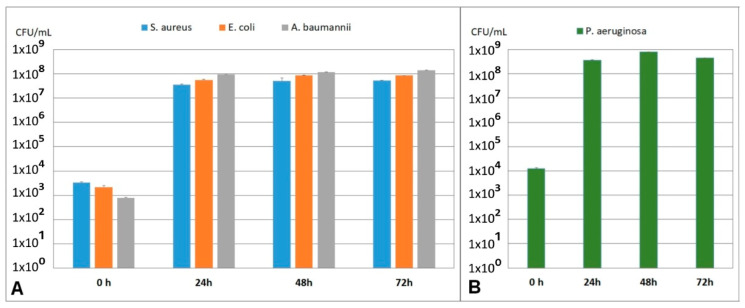
Stability testing of the multispecies 3D biofilm model (**A**) with *S. aureus*, *E. coli*, and *A. baumannii*, and the *P. aeruginosa* 3D biofilm model (**B**). After 24 h, a stable microbial population is established in both models, which remains stable until 72 h. The ratio of the bacteria to each other in the multispecies 3D biofilm model (**A**) remains the same during this period.

**Figure 5 microorganisms-10-02027-f005:**
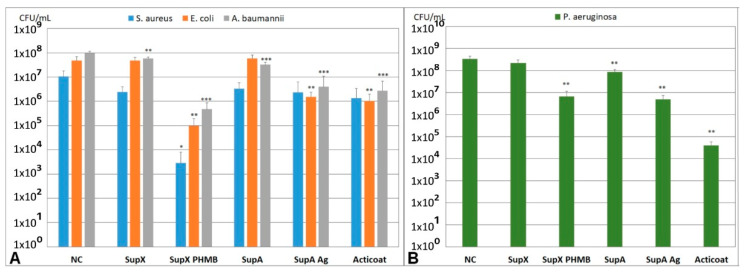
Antimicrobial effect of 24 h incubation of the multispecies 3D biofilm model with *S. aureus*, *E. coli*, and *A. baumannii* (**A**), and the 3D biofilm model with *P. aeruginosa* (**B**) with wound dressings. Asterisks indicate significant deviations from the untreated negative control: * *p* < 0.05; ** *p* < 0.01; *** *p* < 0.001.

## Data Availability

All outcome data are available as representative images in the main text. The raw datasets generated and analyzed during the current study are available from the corresponding author on reasonable request.

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
