# Peer review of "3D Biofilm Models Containing Multiple Species for Antimicrobial Testing of Wound Dressings"

_microorganisms, 2022, doi:10.3390/microorganisms10102027_

Round 1
Reviewer 1 Report
The authors report a novel 3D laboratory model for bacterial biofilms that creates multispecies biofilms in "bubbles" formed in a mixed solution of biological agar and gelatin in nutrient rich medium. They provide good microscopic evidence of mature 48-hour multispecies biofilms formed from S. aureus, E. coli and A. baumannii within the 'bubbles' or single species P. aeruginosa, which that remain stable for up to 72 hours. They also provide data on quantification of the antimicrobial effect of multiple wound dressings using the 3D model.
The authors need to address the following questions.
1. The authors need to include one additional assessment of the new 3D biofilm model, which is to measure the effectiveness of exposing the 48-hour 3D biofilms to 24 hours of 50X the minimum bactericidal concentration (MBC) of appropriate antibiotic(s) then measuring the CFUs of surviving viable bacteria. This is because some of the "biofilm bacteria" will not be tolerant to the antibiotic due to retaining some metabolic activity that will make those bacteria susceptible to killing by the antibiotics. For example, if 24-hours of exposure of the 48-hour 3D biofilm model to 50X MBC of gentamycin reduced the total CFUs by 1-log or 2-logs then the effects of the antimicrobial wound dressings on ONLY MATURE TOLERANT BIOFILM BACTERIA would be 10-times or 100-times less than the log reduction of the "BIOBURDEN" that included the killing (reduction) of TOTAL bacteria CFUs. Also, this information would help readers understand the effects of the different wound dressings on the planktonic bacteria population and the biofilm bacteria population. The authors may even include a new paragraph in the Discussion section #4 that researchers may want to modify the 3D model protocol to include placing the 48-hour 3D biofilm model samples into an antibiotic containing medium for 24 hours before exposing the 3D biofilm model samples to the test wound dressings for 24 hours so the resulting CFUs measured would reflect killing of ONLY antibiotic tolerant bacteria in the mature 3D model. This is similar the procedure that was reported by P.L. Phillips, et al, Effects of Antimicrobial Agents on an In Vitro Biofilm Model of Skin Wounds, Advances Wound Care, 1: 299-304, 2010., who assessed the effects of wound dressings on mature bacterial biofilms grown on the dermis of pig skin explants.
2. The authors should include a statement in the Methods section that explained what steps were taken to neutralize any residual antimicrobial agents (chemicals) that were released from the wound dressings into the agar/gelatin matrix during the 24 hour exposure and that would subsequently be solubilized into plating medium during the during the homogenization of the 3D biofilm matrix. The antimicrobial agents that are release during the homogenization would be able to kill the biofilm bacteria that were dispersed into single bacteria or microcolonies of bacteria that were no longer protected because the biofilm matrix was disrupted during the homogenization process before plating.
3. Please change the color of the "light green" bars in panel B of Figure 5 to a much darker green color (or a different color) because the bars are difficult for readers to easily distinguish from the white color background.
4. Please enlarge the font size of the labels for the test conditions presented along the X-axis in panel A of Figure 5 because it is VERY DIFFICULT for readers to easily read the tiny letters in the current figure.
a 24-hour incubation period in antibiotic medium of
Reviewer 2 Report
Comments on Reddersen et al:
The aim of this manuscript is to establish 3D biofilm models, for antimicrobial testing of wound dressing, with multispecies communities of bacteria, localized in chronic wounds.
This manuscript shows rich content, providing a deep insight for some works: the study is within the journal’s scope, and I found it to be well-written, providing sufficient information. Even if the manuscript provides an organic overview, with a densely organized structure and based on well-synthetized evidence, there are some suggestions necessary to make the article complete and fully readable. For these reasons, the manuscript requires minor changes.
Please find below an enumerated list of comments on my review of the manuscript:
INTRODUCTION:
LINE 28: This type of microbial lifestyle provides the microorganism with protection from physical, chemical and biological hazards; at the same time, biofilm are mainly involved in several infective disease, which often require preventive strategies (see, for reference: Bernardi, S.; Anderson, A.; Macchiarelli, G.; Hellwig, E.; Cieplik, F.; Vach, K.; Al-Ahmad, A. Subinhibitory Antibiotic Concentrations Enhance Biofilm Formation of Clinical Enterococcus faecalis Isolates. Antibiotics 2021, 10, 87 https://doi.org/10.3390/antibiotics10070874) In this introductive section, the authors should describe the underlying dynamics of microbial community coexistence and biofilm involvement, in several infectious disease.
LINE 38 – 41: Furthermore, biofilm formation is triggered and regulated by several variables: from environmental to hydrodynamic conditions, even if also nutrient availability and cell signaling might be deeply involved in this process, as suggested by several and recent studies (see, for reference: Rather MA, Gupta K, Mandal M. Microbial biofilm: formation, architecture, antibiotic resistance, and control strategies. Braz J Microbiol. 2021 Dec;52(4):1701-1718. doi: 10.1007/s42770-021-00624-x. Epub 2021 Sep 23. PMID: 34558029; PMCID: PMC8578483).
The main topic is interesting, and certainly of great clinical impact. As regards the originality and strengths of this manuscript, this is a significant contribute to the ongoing research on this topic, as it extends the research field on the development of 3D biofilm models, for antimicrobial testing of wound dressing, with multispecies communities of bacteria, localized in chronic wounds. Overall, the contents are rich, and the authors also give their deep insight for some works.
As regards the section of methods, there is a specific and detailed explanation for the methods used in this study: this is particularly significant, since the manuscript relies on a multitude of methodological and statistical analysis, to derive its conclusions. The methodology applied is overall correct, the results are reliable and adequately discussed.
The conclusion of this manuscript is perfectly in line with the main purpose of the paper: the authors have designed and conducted the study properly. As regards the conclusions, they are well written and present an adequate balance between the description of previous findings and the results presented by the authors.
Finally, this manuscript also shows a basic structure, properly divided and looks like very informative on this topic. Furthermore, figures and tables are complete, organized in an organic manner and easy to read.
In conclusion, this manuscript is densely presented and well organized, based on well-synthetized evidence. The authors were lucid in their style of writing, making it easy to read and understand the message, portrayed in the manuscript. Besides, the methodology design was appropriately implemented within the study. However, many of the topics are very concisely covered. This manuscript provided a comprehensive analysis of current knowledge in this field. Moreover, this research has futuristic importance and could be potential for future research. However, minor concerns of this manuscript are with the introductive section: for these reasons, I have minor comments only for the introductive section, for improvement before acceptance for publication. The article is accurate and provides relevant information on the topic and I have some minor points to make, that may help to improve the quality of the current manuscript and maximize its scientific impact. I would accept this manuscript if the comments are addressed properly.
Reviewer 3 Report
The manuscript tried to describe a 3D model for biofilms. But, only some preliminary experimental results were reported. The comparison with other models or clinical data was totally missed. I even can not be convinced that what authors prepared is a model. The problems they described in the introduction were not sufficienctly revealed and resolved correspondingly. As a result, I can not find any solid novelty. So, it must be rejected at current form.
Round 2
Reviewer 3 Report
I still cannot grasp the novelty without necessary comparisons. The authors presented their own results and did not convince me that they developed a new model more similar with clinical observations. So, I do not think the revised manuscript is worth publication.